# Smoking Has No Influence on Outcomes after Repair of the Medial Meniscus in the Hypo and Avascular Zones—A Pilot Study

**DOI:** 10.3390/ijerph192316127

**Published:** 2022-12-02

**Authors:** Jan Zabrzyński, Łukasz Paczesny, Agnieszka Zabrzyńska, Gazi Huri, Kamil Graboń, Tomasz Pielak, Jacek Kruczyński, Łukasz Łapaj

**Affiliations:** 1Department of General Orthopaedics, Musculoskeletal Oncology and Trauma Surgery, University of Medical Sciences, 61-545 Poznan, Poland; 2Department of Orthopaedics, Orvit Clinic, Citomed Healthcare Center, 87-100 Torun, Poland; 3Faculty of Medicine, Collegium Medicum in Bydgoszcz, Nicolaus Copernicus University in Toruń, 85-092 Bydgoszcz, Poland; 4Department of Radiology, Multidisciplinary Hospital, 88-100 Inowroclaw, Poland; 5Orthopaedics and Traumatology Departament, Hacettepe University School of Medicine, Ankara 06-230, Turkey; 6Department of Orthopaedics, Clinical Hospital, 25-736 Kielce, Poland

**Keywords:** meniscus, smoking, all-inside suture, meniscus tear, meniscus suture, meniscus repair, knee, arthroscopy

## Abstract

Complete loss of the meniscus inevitably leads to knee joint degeneration. Smoking is an important factor predicting poor outcome in orthopedics; however, data about its role in meniscus surgery are inconclusive. Smoking could be an important negative factor in isolated meniscus repair. The aim of this paper was to determine the influence of smoking on functional outcomes after isolated all-inside medial meniscus repair. This study included 50 consecutive patients with isolated, traumatic tear of the medial meniscus who underwent knee joint arthroscopy between 2016 and 2019. All-inside arthroscopic repair of the medial meniscus was performed in each case. All patients followed a uniform, postoperative rehabilitation protocol for 8 weeks. The follow-up examination was based on the functional scores at 3 and 6 months postoperatively. According to smoking status there were 17 smokers and 33 non-smokers. The mean number of cigarettes smoked per day was 11, for a mean of 7.4 years, and the mean pack-years index value was 4.9. There was no correlation between smoking years, number of cigarettes smoked per day, pack-years index, and functional outcomes. The arthroscopic inspection of the knee joints revealed cartilage lesions (≤IIº) in eight subjects, suggesting the secondary pathology to the meniscus tear. In this study, we found no evidence of an association between smoking indices and functional outcomes after all-inside repair of chronic medial meniscus tear. The nature of the chronic meniscal tear could be smoking-resistant owing to the poor blood supply to the sites in which these specific lesions occur.

## 1. Introduction

Historically, menisci were often excised with an open total meniscectomy, because it was believed that they have no functional meaning [1]. Nowadays, it has been proved that menisci have multiple functions in the knee joint, e.g., they are involved in load transmission, shock absorption, proprioception, and help in articular cartilage nutrition, lubrication, and protection [1,2,3,4]. The complete loss of a meniscus or partial defect unavoidably lead to cartilage damage and further knee joint degeneration [5]. Recent studies revealed that the postoperative function of the knee is linked to the size and the structure of the preserved meniscus. Thus, awareness of the serious consequences of meniscus loss has directed surgeons toward joint preservation techniques and mini-invasive surgery to the new paradigm, “save the meniscus” [3]. Novel methods of treatment, such as meniscal repairs, meniscal allografts, and scaffolds, avoid the impact of meniscectomy in terms of knee function and of progression toward osteoarthritis [1,2,3,6,7,8].

Injuries to the menisci are the second most common trauma to the knee, with a prevalence of 12% to 14%, and data are still lacking on the epidemiology and prognostic factors of meniscal injuries [8,9]. During joint motion, the meniscus has to resist various forces: tension, compression, and shear stress [10]. Snoeker et al. suggested that squatting, kneeling, crawling, chair sitting, stair climbing, lifting, and walking were all risk factors for meniscal tears [11]. The link between cartilage lesions and concurrent meniscus tears is known as important negative prognostic factor in anterior cruciate ligament (ACL) surgery and leads to impaired functional scores, as reported in many papers [12,13]. Meniscal tears were also found to be associated with increased cartilage loss in the involved compartment [14]. Smoking is an important negative factor in orthopedic surgery; however, the positive effect of nicotine on chondrocyte metabolism has been proposed by in vitro studies [15]. In a few studies, smoking was under consideration as one of the main factors affecting the surgery of the meniscus; however, there are also papers that clearly show the absence of a role [11,13]. 

This study was conceived owing to inconclusive data about the effect of smoking on clinical results after meniscus surgical repair. Mostly, authors deal with complex knee trauma, with a mix of ACL, medial meniscus (MM), lateral meniscus (ML) injuries, and cartilage defects. Smoking may be an important negative factor in isolated meniscus repair. The aim of this paper was to examine the influence of smoking on functional outcomes after isolated all-inside medial meniscus repair.

## 2. Materials and Methods

### 2.1. Patients Selection and Preoperative Assessment

This study included 50 consecutive patients with isolated, traumatic tear of MM, who underwent knee joint arthroscopy at Department of Orthopaedics between 2016 and 2019, performed by two surgeons. The study was performed following the Declaration of Helsinki for experiments involving humans, after receiving permission from the local Bioethics Committee (approval number 29/KB/2016). All patients provided written informed consent before participating in the study. 

The inclusion criteria were medial meniscus tear suspicion established on typical clinical symptoms, a positive McMurray’s/Appley’s/Childress’ tests, presence of isolated MM rupture diagnosed preoperatively in magnetic resonance imaging (MRI, 1.5 Tesla), confirmed arthroscopically and further all-inside arthroscopic meniscus repair [16].

The exclusion criteria were positive Lahman’s/Drawer/Pivot shift tests in clinical examination and the presence of intra-articular, concomitant pathology confirmed in MRI after trauma, e.g., ACL tear, lateral meniscus tears, cartilage lesions > Iº International Cartilage Repair Society Score (ICRS), previous surgery of the affected knee joint, the period between trauma and surgery of >3 months, and loss of contact with subject during the follow-up.

Preoperative data about age, gender, sport activity, Lysholm knee scoring system (LKSS), and International Knee Documentation Committee (IKDC) scores had been recorded. Patients were asked about their actual smoking status and smoking habits in the past. The population was divided into smokers (did not give up habit before surgery and after surgery) and non-smokers (did not consume tobacco products for their whole life and during the postoperative period). We also recorded the parameters of cigarette smoking—smoking years, the mean number of cigarettes smoked per day, and pack-years index (1 pack has 20 cigarettes).

### 2.2. Surgical Treatment

Patients were positioned supine in a manner typical of knee arthroscopy, a standard 30-degree 5.6 mm arthroscope was inserted through standard arthroscopic portals. All knee joint compartments were explored and the MM injury was checked with the arthroscopic probe. Meniscus injuries were classified by size, location, and morphology of tears. If a preoperative diagnosis was confirmed, then an appropriate procedure was performed. Since the repair in avascular zone is not a contraindication for repair, the surgery was also performed in this area of meniscus [17]. The all-inside repair was performed by absorbable vertical sutures with the use of disposable needles and the number of used sutures was noted depending on tears morphology. Medial collateral ligament pie-crusting was routinely performed for medial compartment visualization and easier surgical technique. Cartilage was assessed by probe palpation and arthroscopic evaluation; if recognized, the coexisting cartilage degeneration was registered in line with the ICRS classification. 

### 2.3. Rehabilitation Protocol

All patients followed the same postoperative rehabilitation protocol. The knee was kept in a brace, for a period of 6 weeks, with an ROM limitation of 90 degrees for 4 weeks. Partial weightbearing was allowed with the assistance of crutches with progression as tolerated. Gradual progress of range of motion beyond 90 degrees after 4 weeks. Progression to full weightbearing after 4 weeks. Functional rehabilitation after 6 weeks with aim to regain full range of motion after 8 to 12 weeks.

### 2.4. Outcomes Assessment

The follow-up examination included outcomes assessment by physical examination, the LKSS and the IKDC scores at 3 and 6 months postoperatively, clinical tests dedicated for meniscus pathology used preoperatively, current smoking status, occurrence of complications, and failures with re-operations.

### 2.5. Statistical Analysis

Variables were tested for normality by Shapiro–Wilk test. Correlation of variables within patients were conducted with the Spearman Rho correlation coefficient. The Mann–Whitney U test and the Wilcoxon test were used for statistical comparison of independent and dependent data, respectively. The Chi squared test was used to compare descriptive characteristics. All the comparison between groups and statistical analysis were performed blinded and by independent investigator. We used IBM SPSS Statistics v.25 (IBM Corp., Armonk, NY, USA) to perform all statistical analyses. A *p*-value < 0.05 was considered to be statistically significant.

## 3. Results

The mean age in the studied group was 41.68 years (range: 13–66; SD = 12.23) on enrollment; the gender distribution was 18 women to 32 men (Table 1).

By smoking status, there were 17 smokers and 33 non-smokers. Specifically, the mean age was 42.05 (range 17–61 SD = 13.03) in the smokers group and 41.48 (range 13–66 SD = 12.01) in the non-smokers group (*p* = 0.8138). The average number of cigarettes smoked per day was 11.29 (range: 2–30 SD = 8.19), and the mean period of smoking was 7.47 years (range 2–20 SD = 4.96). The mean pack-years index was 4.9 (range 0.2–20 SD = 5.25). No patients changed their smoking habit during the follow-up. All patients in the cohort had a positive clinical test for medial meniscus pathology and a positive history of trauma. The arthroscopic inspection of knee joints confirmed MM tears in each case and additionally revealed intra-operative finding of cartilage lesions in eight subjects (all ≤IIº ICRS), which were found in the preoperative MRI scans, suggesting it is a secondary process to meniscus tears. The mean number of all-inside sutures used in meniscal repairs was 3 (range 2–7; SD = 1.2). Thirty-seven patients had a simple meniscus tear (longitudinal, radial, or horizontal) and thirteen had a complex medial meniscus tear (Figure 1). 

No patient was excluded during the follow-up period. There was no re-operation during the follow-up and no complications occurred. The Chi squared test was used to compare descriptive characteristics: age > 45 or <45, number of sutures ≤ 2 or >2, simple or complex tears, LKSS after 6 months < 83 or ≥83, IKDC after 6 months < 83 or ≥83, cartilage defect absent or present, in the smokers and non-smokers groups (Table 2).

All patients displayed a significant improvement in their clinical outcome measured in LKSS; from 62.62 (range 12–89; SD = 18.63) preoperatively to 84.14 (range 56–99; SD = 10.12; *p* < 0.0001) after 3 months and 90.16 (range 69–100; SD = 8; *p* < 0.0001) after 6 months postoperatively. Furthermore, significant improvement in IKDC score was also noted: 47.31 (range 13–82; SD = 16.93) to 71.5 (range 42–94; SD = 12.65; *p* < 0.0001) and 80.8 (range 43–98; SD= 12.19; *p* < 0.0001), respectively. In the group of non-smokers, the outcomes increased from 66.6 (range 28–84; SD = 15.79) preoperatively to 83.81 (range 56–99; SD = 10.29; *p* < 0.0001) after 3 months and 90.96 (range 79–100; SD = 7.54; *p* < 0.0001) after 6 months postoperatively for LKSS. The improvement was also noted in IKDC score: 49.01 (range 20–80; SD = 16.38) to 70.53 (range 42–90; SD = 13.67 *p* < 0.0001) and 81.18 (range 47–98; SD = 12.01 *p* < 0.0001), respectively. The smokers group had an improved LKSS score from 54.88 (range 12–89; SD = 21.63) preoperatively to 84.76 (range 60–99; SD = 10.07 *p* = 0.0001) after 3 months and 88.58 (range 69–100; SD = 8.84; *p* > 0.05) after 6 months postoperatively. On the other hand, an improvement in IKDC score was also noted: 44.01 (range 13–82; SD = 17.99) to 73.38 (range 48–94; SD = 10.52, *p* = 0.0001) and 81.73 (range 43–98; SD = 12.88, *p* = 0.029), respectively.

There was no correlation between smoking-years, number of cigarettes smoked per day, pack-year index, and functional outcomes (Table 3).

## 4. Discussion

We found no evidence of an association between smoking indices and functional outcomes after all-inside repair of chronic medial meniscus tear. To discuss and explain this fact, we should start from basic science. The vascularization and nutritional situation of a torn meniscus area, as well as the morphological type of rupture, are crucial for the success of a meniscus surgery [2]. The blood supply to the menisci is very important implication to the potential healing. Blood vessels mainly originate from the medial and lateral geniculate arteries; however, only the peripheral 10–25% of the meniscus benefits from blood supply in the mature skeleton, in a so-called red-red zone [2]. The inner one-third of the meniscus, the so-called white zone, is supplied by diffusion from the synovial fluid, while the transition red-white region in the middle has attributes of each zone. Former recommendations advised suturing menisci only in the red-red zone, but Barber-Westin et al. proved that the meniscus also had regenerative potential in hypovascular zone, with an 86% healing rate for menisci in the red-white zone [18]. Tobacco smoke and its components such as carbon monoxide impede vascularization, decreasing microperfusion and tissue oxygenation [19]. The toxic dose of smoke and its dose-dependent effects on fibrocartilaginous tissue of meniscus are still unknown. It appears to be clear that tobacco smoke impairs the regeneration of meniscus and subsequently the functional outcomes; however, data from our study and others do not support this [11,20]. The morphology of tears in our cohort appeared as simple or complex, but usually occupied hypo or avascular zones. Those so called “degenerative” tears could be resistant to smoke influence due to physiologically impaired vascularization. Moreover, we did not note complications during the follow-up, such as re-rupture, so the meniscus structure seems to be healed, but this may probably result also from multiple sutures used in surgical technique.

There are only few studies that consider the impact of smoking on meniscus lesions. In our study there was no link between smoking indices and functional outcomes. Snoeker et al. also showed no association of smoking status and meniscal tears in their meta-analysis [11]. Laurendon et al. analyzed 87 patients after all-inside meniscus repair and, similar to our results, did not find smoking habit as a statistically important factor [20]. In contrast, Blackwhell et al. revealed that current smokers were significantly more likely to undergo secondary meniscectomy in mid-term follow-up after the primary meniscus repair however, compared with our study, they have longer follow-up and most subjects underwent concomitant ACL surgery, which could bias the outcomes [14]. Moreover, authors who noted the negative impact of smoking habit referred mostly to complex knee joint injuries usually with concurrent pathologies [13,21,22]. We decided to carefully select patients included into the study to isolate subjects with medial meniscus lesions. Additionally, smokers in our study had relatively low smoking indices that may have tended to produce good outcomes. Authors in other studies focused only on smoking status and did not consider the mentioned indices [13,22].

The prevalence of knee cartilage lesions is about 60–70% during arthroscopy and the most frequent concomitant injuries are medial meniscus ruptures and ACL tears [20]. Laurendon et al. did not find cartilage injuries or smoking habit as a statistically important threat for arthroscopic meniscal repair [20]. Compared with our study, their group was younger and the follow-up was longer; moreover, they focused on the failure of meniscal repair that demanded a secondary meniscectomy. Despite the older population, we did not observe failure of MM isolated repair; however, the mean number of applied sutures in our study was 3 vs. 1.73 used in the study of Laurendon et al. [20]. Despite the lack of chondral lesions in a rapid post-injury MRI, we encountered them during intra-articular inspection; however, it was ≤IIº according to ICRS. Cartilage lesions must have appeared during the preoperative period, probably due to MM tears and subsequent cartilage damage. Interestingly, these injuries were more frequently found in patients < 45 years. It is probably connected with sports activity and the force of the primary injury. Cox et al. showed that cartilage trauma impairs clinical results after meniscus surgery; specifically, the greater the damage to the cartilage and meniscus, the worse the clinical outcomes [2,21]. Webster et al. revealed that chondral and meniscal injuries significantly decrease the outcomes following revision of the ACL reconstruction. In particular, the medial meniscus negatively influenced functional scores, in contrast to lateral meniscus and cartilage defects more than IIIº, according to ICRS, significantly impaired the results [23]. In the present study, we did not confirm such serious cartilage lesions, even though they also had an impact on clinical outcomes. A negative correlation was revealed 6 months after surgery between IKDC score and chondral lesions.

Uzun et al. did not find a link between age and clinical outcomes [22]. We also did not find statistically significant differences between the >45 and <45 years groups. Weber et al. revealed that the healing rate of meniscus suturing increases the sooner it is performed, optimally <12 weeks after trauma, and it was a cut-off point in our study inclusion criteria. This could be the cause of good functional outcomes in both smokers and non-smokers group. Moreover, the cartilage defects were limited to IIº according to ICRS, but a longer time from trauma to surgery could lead to more advanced chondral lesions.

The main limitations of this study were the modest number of included patients, various demographic data and morphological types of medial meniscus tears; thus, to create a more coherent population we set strict inclusion criteria. However, we realize that the more coherent studied population, the greater the statistical power. We tried to exclude patients with cartilage injuries; however, in the initial MRI scan, the cartilage defects found during knee arthroscopy had low grades according to ICRS and probably these defects were secondary to meniscus lesions. The next limitation was lack of structural imaging (MRI) to confirm the status of the articular cartilage and meniscus at final follow-up. Low-grade cartilage lesions could be filled by fibrocartilaginous tissue. However, good clinical outcomes indirectly suggested medial menisci regeneration. The longer follow-up could show the aftermath of injury, such as re-operation rate or degenerative changes in the knee joint. Furthermore, we could not obtain reliable information about the nicotine concentration and other harmful substances that all subjects had consumed; thus, dose-dependency cannot be estimated from this study. The follow-up was relatively short, but authors revealed that secondary arthroscopy showed that the meniscus tear healed within 3 months after primary surgical repair and could be used a threshold for major failures after meniscal surgery [7].

## 5. Conclusions

Meniscus surgery outcome is closely linked to vascularization pattern. In this study, we found no evidence of an association between smoking indices and functional outcomes after all-inside repair of the chronic medial meniscus tear. The nature of chronic, so-called degenerative meniscal tears could be smoking-resistant due to the poor blood supply to the sites in which these specific lesions occur.

## Figures and Tables

**Figure 1 ijerph-19-16127-f001:**
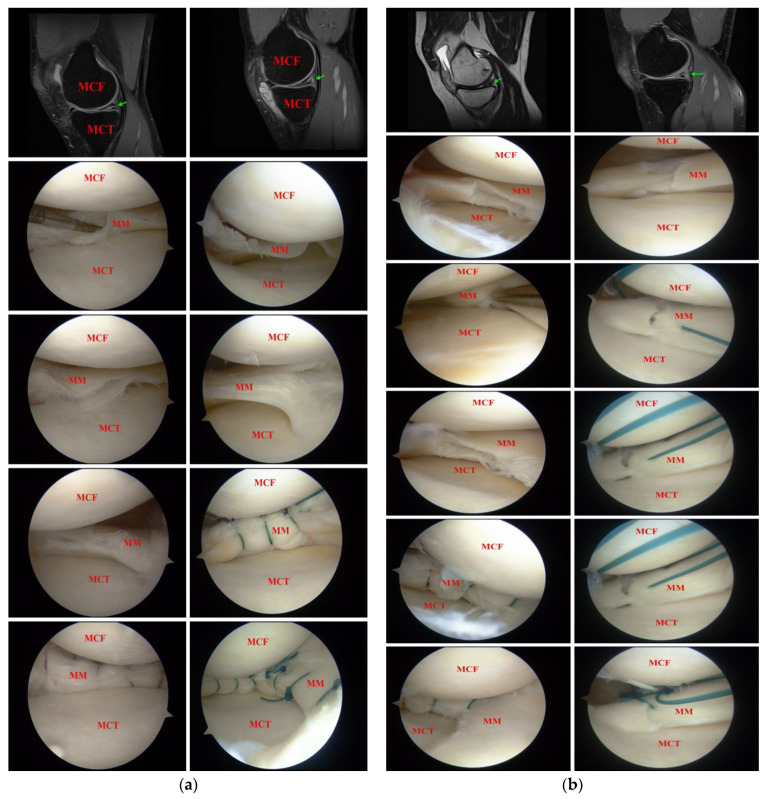
Arrows indicate the medial meniscus tears in the MRI scans (**a**,**b**). (**a**,**b**) present the subsequent steps of all-inside repair of the MM, all were morphologically cleaved meniscal tears with some complex component, and were repaired with the same technique. This all-inside technique is performed using disposable needles, suture grasper, with a few horizontal/vertical all-inside sutures around the meniscus body. MM—medial meniscus, MCF—medial femoral condyle, MCT—medial tibial condyle.

**Table 1 ijerph-19-16127-t001:** Summary of demographic and clinical characteristics.

Variables	Total	Smokers	Non-Smokers
Population	n = 50	n = 17	n = 33
Mean age (years)	41.68 ± 12.23	42.05 ± 13.03	41.48 ± 12.01
Gender			
Male	32	12	20
Female	18	5	13
Mean duration of smoking (years)	-	7.47 ± 4.96	-
Mean number of cigarettes per day	-	11.29 ± 8.19	-
Pack-years index	-	4.9 ± 5.25	-
LKSS preop.	62.62 ± 18.63	54.88 ± 21.63	66.6 ± 15.79
LKSS 3 m. postop.	84.14 ± 10.12	84.76 ± 10.07	83.81 ± 10.29
LKSS 6 m. postop.	90.16 ± 8	88.58 ± 8.84	90.96 ± 7.54
IKDC preop.	47.31 ± 16.93	44.01 ± 17.99	49.01 ± 16.38
IKDC 3 m. postop.	71.5 ± 12.65	73.38 ± 10.52	70.53 ± 13.67
IKDC 6 m. postop.	80.8 ± 12.19	81.73 ± 12.88	81.18 ± 12.01

**Table 2 ijerph-19-16127-t002:** Summary of the descriptive characteristics among patients who underwent medial meniscus surgery.

	Smokers	Non-Smokers	*p* Value *
**Age**			
<45	10	22	0.8132
>45	7	11	
**Tear Morphology**			
Simple	6	17	0.4291
Complex	11	16	
**No. of Sutures**			
≤2	5	7	0.7691
>2	12	26	
**LKSS 6 m.**			
<83	3	6	0.7324
≥83	14	27	
**IKDC 6 m.**			
<83	11	16	0.4291
≥83	6	17	
**Cartilage defect**			
No defect	12	30	0.1472
Defect present	5	3	

* Chi Square-test; Statistically significant (*p* < 0.05). Legend: 

 0–9 patients; 
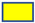
 10–19 patients; 
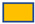
 20–29 patients; 

 ≥30 patients.

**Table 3 ijerph-19-16127-t003:** Statistical correlations between functional outcomes (LKSS, IKDC) and various factors.

	LKSS Preop.	LKSS 3 m. Postop.	LKSS 6 m. Postop.	IKDC Preop.	IKDC 3 m. Postop.	IKDC 6 m. Postop.
Smoking-years	0.1524	0.4946	0.7230	0.2003	0.7792	0.8711
No. of cigarettes	0.1079	0.4360	0.6058	0.4488	0.6409	0.7489
Pack-year index	0.1458	0.4326	0.6600	0.3421	0.7333	0.8058
Age	0.0987	0.7985	0.9918	0.1468	0.3733	0.1027

## Data Availability

Not applicable.

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
