# Peer review of "Smoking Has No Influence on Outcomes after Repair of the Medial Meniscus in the Hypo and Avascular Zones—A Pilot Study"

_ijerph, 2022, doi:10.3390/ijerph192316127_

Round 1

Reviewer 1 Report

Title: ok

Abstract: hypothesis is missing

do not use abbreviations in the abstract

conclusions: ok

Introduction is really long

please report controversies about the topic, clear ratio for the study and what is new and then aim and hypothesis

methods: strobe checklist are missing

was a power analysis performed to ensure sample size?

results are really well written and also images are ok

discussion is a little bit long

start with the main findings of the study

what this add to current literature? what is new?which are the clinical implications of the study?

conclusions coherent

references:

please add following references: 

D'Ambrosi R, Meena A, Raj A, Ursino N, Mangiavini L, Herbort M, Fink C. In elite athletes with meniscal injuries, always repair the lateral, think about the medial! A systematic review. Knee Surg Sports Traumatol Arthrosc. 2022 Nov 2. doi: 10.1007/s00167-022-07208-8. Epub ahead of print. PMID: 36319751.

Author Response

Dear Editor,

Thank you for the opportunity to improve and resubmit our manuscript entitled:

“Smoking has no Influence on The Outcomes after Repair of The Medial Meniscus in the Hypo and Avascular Zones – a pilot study”

The suggestions offered by the reviewers have been immensely helpful. We appreciate the clarification of the concerns about the paper.

We have included the reviewer comments, and responded to them individually, indicating how we addressed each concern and describing the changes we have made:

Title: ok

Abstract: hypothesis is missing – it was added

do not use abbreviations in the abstract – it was corrected

conclusions: ok

Introduction is really long

please report controversies about the topic, clear ratio for the study and what is new and then aim and hypothesis – it was revised.

methods: strobe checklist are missing – it was added

was a power analysis performed to ensure sample size? – no, it wasn’t

results are really well written and also images are ok  - thank you

discussion is a little bit long - revised

start with the main findings of the study - it was added

what this add to current literature? what is new?which are the clinical implications of the study? – it was revised

conclusions coherent - thanks

references:

please add following references: - consider and done.

We wish to express again our appreciation for the insightful comments which have helped us significantly to improve our manuscript.

Reviewer 2 Report

The aim of the present article is to evaluate the occurrence of an association between smoking indexes and functional outcomes after all-inside repair of the chronic medial meniscus tear. 

There are some minor issues, which are reported below, that need to be addressed or fixed.

Abstract: please avoid the use of abbreviations in the abstract (lines 21 and 26).

Lines 28-30. the Authors should consider to better detail this aspect, which may be crucial for data interpretation.

Introduction:

Line 48. Please remove the underscore

Lines 51-53: please add some references

Lines 57, 67, 69, line 72: Please revise the use of abbreviation. 

Results: Line 151 should be moved in the legend of figure 1. The authors should better detail in the figure legend and/or in the figure itself the subsequent steps; which is the difference between pictures in A and B? this is not very clear to the reader.

Discussion:

Lines 202-203: “Moreover, we did not noted complications”-- please change “noted” into “note”

Lines 234-236: Please revise this sentence, since it is not clear to the reader.

Line 253: “criteria, however”. Please add a comma between criteria and however.

Lines 254-256: Please revise this sentence, since it is not clear to the reader.

Author Response

Dear Editor,

Thank you for the opportunity to improve and resubmit our manuscript entitled:

“Smoking has no Influence on The Outcomes after Repair of The Medial Meniscus in the Hypo and Avascular Zones – a pilot study”

The suggestions offered by the reviewers have been immensely helpful. We appreciate the clarification of the concerns about the paper.

We have included the reviewer comments, and responded to them individually, indicating how we addressed each concern and describing the changes we have made:

The aim of the present article is to evaluate the occurrence of an association between smoking indexes and functional outcomes after all-inside repair of the chronic medial meniscus tear. 

There are some minor issues, which are reported below, that need to be addressed or fixed.

Abstract: please avoid the use of abbreviations in the abstract (lines 21 and 26). – revised.

Lines 28-30. the Authors should consider to better detail this aspect, which may be crucial for data interpretation. – we need to adapt to publisher criteria of the Abstract length, that is why the M&M section in the Abstract section is rather concise.

Introduction:

Line 48. Please remove the underscore – it was removed

Lines 51-53: please add some references – it was added

Lines 57, 67, 69, line 72: Please revise the use of abbreviation. - revised

Results: Line 151 should be moved in the legend of figure 1. The authors should better detail in the figure legend and/or in the figure itself the subsequent steps; which is the difference between pictures in A and B? this is not very clear to the reader. – it was revised.

Discussion:

Lines 202-203: “Moreover, we did not noted complications”-- please change “noted” into “note” – it was revised

Lines 234-236: Please revise this sentence, since it is not clear to the reader. - revised

Line 253: “criteria, however”. Please add a comma between criteria and however. - revised

Lines 254-256: Please revise this sentence, since it is not clear to the reader. – revised.

We wish to express again our appreciation for the insightful comments which have helped us significantly to improve our manuscript.